# Longitudinal Variational Autoencoder for Compositional Data Analysis

**Mine Öğretir** [1]  **Harri Lähdesmäki** [1]  **Jamie Norton** [2]

## Abstract

The analysis of compositional longitudinal data, particularly in microbiome time-series, is a challenging task due to its high-dimensional, sparse, and compositional nature. In this paper, we introduce a novel Gaussian process (GP) prior variational autoencoder for longitudinal data analysis with a multinomial likelihood (MNLVAE) that is specifically designed for compositional time-series analysis. Our generative deep learning model captures complex interactions among microbial taxa while accounting for the compositional structure of the data. We utilize centered log-ratio (CLR) and isometric log-ratio (ILR) transformations to preprocess and transform compositional count data, and utilize a latent multi-output additive GP model to enable prediction of future observations. Our experiments demonstrate that MNLVAE outperforms competing method, offering improved prediction performance across different longitudinal microbiome datasets.

## 1. Introduction

The analysis of compositional longitudinal data, especially in the context of microbiome time-series, has gained increasing attention in recent years. Microbiome studies focus on understanding the complex microbial communities and their interactions within various ecosystems, including the human body. Longitudinal analysis of microbiome data allows researchers to explore temporal patterns, community dynamics, and the impact of various factors on the microbial ecosystem. However, analyzing high-dimensional, sparse, and compositional count data from microbiome time-series presents significant challenges.

Early works on analyzing microbiome dynamics ignored the compositional nature and analyzed the temporal profiles of individual microbes or strains separately, e.g. (Kostic et al., 2015; Vatanen et al., 2016; 2019). Äijö et al. (2018) proposed a Bayesian non-parametric method for compositional analysis of microbiome time-series using multi-output Gaussian process (GP) together with the multinomial likelihood but allowed accounting only for time covariate. More recently, recurrent neural network (RNN) based machine learning methods (Metwally et al., 2019; Sharma & Xu, 2021; Chen et al., 2021) as well as differential equation based models, such as the generalized Lotka-Volterra model (Joseph et al., 2020b), have been employed to address these issues.

In this paper, we propose a novel generative model for compositional longitudinal data analysis. The main contributions of our work can be summarized as follows. First, we introduce a new multi-output additive Gaussian process prior VAE with a multinomial likelihood (MNLVAE) that is specifically designed for modeling compositional longitudinal data, such as microbiome time-series analysis, but also applicable to other compositional data domains. Second, our generative deep learning model enables the prediction of future unseen observations, providing valuable insights into microbial community dynamics and potential responses to various factors. Third, by employing a deep latent variable model together with centered log-ratio (CLR) and isometric log-ratio (ILR) transformations, we can effectively capture complex interactions and dependencies among microbial taxa in a more amenable Euclidean space while accounting for the compositional nature of the data. Lastly, we assess the performance of our proposed method on real-world microbiome datasets, demonstrating its performance over an existing approach in terms of predictive capabilities.

## 2. Method

**Notation:** Consider $P$ as the total number of distinct instances (such as individuals), with each instance $p$ containing $n_p$ time-series samples. The complete set of longitudinal samples is given by $N = \sum_{p=1}^{P} n_p$. For each individual $p$, the time-series data comprises a pair $X_p = [\boldsymbol{x}_1^p, \ldots, \boldsymbol{x}_{n_p}^p]^T$ and $Y_p = [\boldsymbol{y}_1^p, \ldots, \boldsymbol{y}_{n_p}^p]^T$, where $\boldsymbol{x}_t^p \in \mathcal{X}$ signifies auxiliary covariate data, and $\boldsymbol{y}_t^p \in \mathcal{Y}$ denotes dependent count variables. The longitudinal data is represented as

[1]Department of Computer Science, Aalto University, Espoo, Finland [2]Integrated Omics AI, United States. Correspondence to: Mine Öğretir <mine.ogretir1@aalto.fi>, Harri Lähdesmäki <harri.lahdesmaki@aalto.fi>.

*Workshop on Interpretable ML in Healthcare at International Conference on Machine Learning (ICML)*, Honolulu, Hawaii, USA. 2023. Copyright 2023 by the author(s).

$(X, Y)$, with $X = [X_1^T, \ldots, X_P^T]^T = [\boldsymbol{x}_1, \ldots, \boldsymbol{x}_N]^T$ and $Y = [Y_1^T, \ldots, Y_P^T]^T = [\boldsymbol{y}_1, \ldots, \boldsymbol{y}_N]^T$.

The domain of $\boldsymbol{x}_n$ is given by $\mathcal{X} = \mathcal{X}_1 \times \ldots \times \mathcal{X}_Q$, where $Q$ represents the total number of auxiliary covariates, and $\mathcal{X}_q$ corresponds to the domain of the $q^{\text{th}}$ covariate, which can be continuous, categorical, or binary. The domain of $\boldsymbol{y}_n = (y_{n1}, \ldots, y_{nD})$ is defined as $\mathcal{Y} = \mathcal{Y}_1 \times \ldots \times \mathcal{Y}_D$, where $\mathcal{Y}_d = \{0, 1, \ldots\}$ for each data dimension $d$ with $D$ specifying the dimensionality of the observed longitudinal count data. The total count of the sample $n$ is $M_n = \sum_{d=1}^{D} y_{nd}$. Finally, the $L$-dimensional latent embedding of $Y$ is denoted by $Z = [\boldsymbol{z}_1, \ldots, \boldsymbol{z}_N]^T = [\bar{\boldsymbol{z}}_1, \ldots, \bar{\boldsymbol{z}}_L] \in \mathbb{R}^{N \times L}$.

## 2.1. Transformations

Aitchison's log-ratio transformations, namely additive log-ratio (ALR), and centered log-ratio (CLR), are widely used in the analysis of compositional data (Aitchison, 1982). These transformations convert compositional data to a Euclidean space while preserving the relative information of the data. The ALR transformation is defined as $\text{ALR}(\boldsymbol{y}) = (\log \frac{y_1}{y_D}, \log \frac{y_2}{y_D}, \ldots, \log \frac{y_{D-1}}{y_D})^T$, where $\boldsymbol{y}$ is a compositional vector of $D$ parts. The CLR transformation is given by $\text{CLR}(\boldsymbol{y}) = (\log \frac{y_1}{g(\boldsymbol{y})}, \log \frac{y_2}{g(\boldsymbol{y})}, \ldots, \log \frac{y_D}{g(\boldsymbol{y})})^T$, where $g(\boldsymbol{y}) = \sqrt[D]{\prod_{i=1}^{D} y_i}$ is the geometric mean of the parts. The ILR transformation utilizes an orthonormal basis matrix, denoted as $\boldsymbol{\Psi}$, and is defined as $\text{ILR}(\boldsymbol{y}) = \boldsymbol{\Psi}\text{CLR}(\boldsymbol{y})$ (Egozcue et al., 2003; Egozcue & Pawlowsky-Glahn, 2005).

In our MNLVAE model, we employed both CLR and ILR transformations to handle compositional data. We utilized the ILR transformation, as it is particularly suited for microbiome data where the phylogenetic tree can be expressed as a binary tree, which in turn can be represented by an orthonormal basis matrix (Morton et al., 2021). For an internal node $l$, the $l$th column vertor of $\boldsymbol{\Psi}$ can be formed as:

$$\boldsymbol{\Psi}_{.l} = (\underbrace{0, \ldots 0}_{k}, \underbrace{a, \ldots a}_{r}, \underbrace{b, \ldots, b}_{s}, \underbrace{0, \ldots, 0}_{t})$$

$$a = \frac{\sqrt{|\boldsymbol{s}|}}{\sqrt{|\boldsymbol{r}|(|\boldsymbol{r}| + |\boldsymbol{s}|)}} \quad b = \frac{-\sqrt{|\boldsymbol{r}|}}{\sqrt{|\boldsymbol{s}|(|\boldsymbol{r}| + |\boldsymbol{s}|)}},$$

where $r$, $s$, $k$ and $t$ are left children, right children, nodes to the left, and nodes to the right of the internal node, respectively. The ILR transformation allows for the efficient encoding of hierarchical relationships among microbial taxa while preserving the original structure of the compositional data.

On the other hand, we also employed the CLR transformation to facilitate comparison and evaluate the performance of our model using different transformation methods. The CLR transformation is a widely used technique in compositional data analysis (Aitchison, 1982) and provides an alternative approach to transforming count data into a more amenable

Euclidean space. By comparing the results obtained with CLR and ILR transformations, we aimed to gain a deeper understanding of the impact of these transformations on the performance of our MNLVAE model.

## 2.2. Longitudinal Variational Autoencoder

Contrary to the classical VAE model, the longitudinal variational autoencoder (L-VAE) model specifies the structure of the data among observed samples by employing an additive multi-output GP prior over the latent space $\boldsymbol{z}|\boldsymbol{x} \sim \mathcal{GP}(\boldsymbol{\mu}(\boldsymbol{x}), k(\boldsymbol{x}, \boldsymbol{x}'|\theta))$, with the prior mean function assumed zero, and the covariance function being modelled using the sum of additive components $k(\boldsymbol{x}, \boldsymbol{x}'|\theta) = \sum_{r=1}^{R} k^{(r)}(\boldsymbol{x}^{(r)}, \boldsymbol{x}'^{(r)}|\theta^{(r)})$, where each component depends on a subset of auxiliary covariates. The GP prior for the latent variables $\boldsymbol{z}$ is assumed to factorize across latent dimensions, but the generative model of L-VAE consists also of a probabilistic decoder $p_\psi(\boldsymbol{y}|\boldsymbol{z})$ that can introduce arbitrary correlations across the latent dimension. The decoder assumes normally distributed data. The KL divergence between the variational posterior of the latent variables and the multi-output additive GP prior can be computed in closed-form, but its exact computation is expensive. An efficient mini-batch compatible upper bound is proposed for the KL divergence that scales linearly to big data, by assuming a standard low-rank inducing point approximation for the covariance function. For a comprehensive understanding of the method and its details, we refer to Section 2.3 and further encourage readers to consult the work by (Ramchandran et al., 2021).

## 2.3. Multinomial Longitudinal Variational Autoencoder

We devised the MNLVAE, drawing inspiration from L-VAE (Ramchandran et al., 2021), and HL-VAE (Öğretir et al., 2022), to effectively handle compositional data by implementing transformations of compositional count data. See Fig. 1 for model overview. To this end, we employed both centered log-ratio (CLR) and isometric log-ratio (ILR) transformations to efficiently process compositional data. The use of ILR was particularly advantageous, as it enabled us to represent phylogenetic trees as binary trees, which can then be expressed using orthonormal basis matrices.

The generative model of MNLVAE is formulated in the following equation:

$$p_\omega(Y \mid X) = \int_Z \underbrace{p_\psi(Y \mid Z, X)}_{\text{Multinomial likelihood}} \underbrace{p_\theta(Z \mid X)}_{\text{GP prior}} dZ$$

$$= \int_Z \prod_{n=1}^{N} p_\psi(\boldsymbol{y}_n \mid \boldsymbol{z}_n) p_\theta(Z \mid X) dZ,$$

where $\omega = \{\psi, \theta\}$ is the set of parameters. The data is

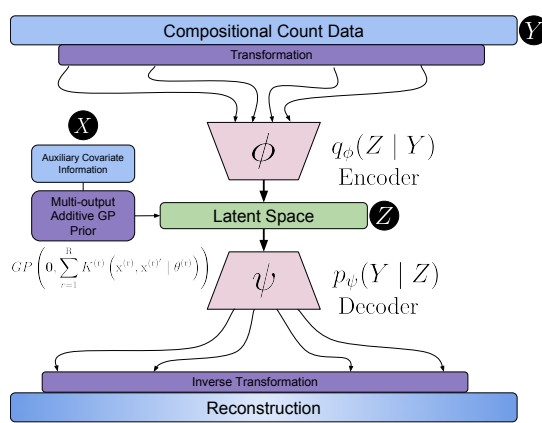

*Figure 1.* Overview of the proposed MNLVAE model.

modelled with a multinomial likelihood:

$$p_\psi\left(\boldsymbol{y}_n \mid \boldsymbol{z}_n\right) = \text{Mult}\left(M_n, \phi(\boldsymbol{\eta}_n)\right)$$
$$\boldsymbol{\eta}_n \mid \boldsymbol{z}_n = f_\psi\left(\boldsymbol{z}_n\right)$$

with the total number of counts $M_n$ for sample $n$. The probability vector $\phi(\boldsymbol{\eta}_n)$ is obtained with the inverse transformation function of the parameters $\boldsymbol{\eta}_n$, where $\phi(\cdot)$ can either be the softmax or the inverse ILR transformation, depending on the chosen transformation technique. The parameters $\boldsymbol{\eta}_n$ is obtained with the decoder function $f_\psi(\boldsymbol{z}_n)$, which is a function of the latent embeddings $\boldsymbol{z}_n$.

We approximate the true posterior of $Z$ variationally using

$$q_\phi(Z \mid Y, X) = \prod_{n=1}^{N} \prod_{l=1}^{L} \mathcal{N}\left(z_{nl} \mid \mu_{\phi,l}\left(\boldsymbol{y}_n\right), \sigma^2_{\phi,l}\left(\boldsymbol{y}_n\right)\right),$$

which is learned together with the generative model parameters via the ELBO objective (see (Ramchandran et al., 2021) for details). Using the trained MNLVAE model with parameters $\phi, \psi, \theta$, we can make future predictions by approximating the predictive distribution. The resulting predictive distribution incorporates covariates associated with unseen test data, and their corresponding latent embeddings as detailed in (Ramchandran et al., 2021).

## 3. Experiments and Results

We utilized three datasets to assess the effectiveness and robustness of our approach for microbiome time-series analysis. We compared our model against the generalized Lotka-Volterra (gLV) model, which is a widely-used ecological model that describes the dynamics of interacting species in a community (Bucci et al., 2016; Stein et al., 2013; Joseph et al., 2020a). The gLV model captures the growth rates, competition, and mutualistic interactions among species, typically represented as a system of ordinary differential equations. It allows for the prediction of species abundance over time, given their initial conditions and interaction parameters, making it suitable for studying various ecological systems and their stability.

### 3.1. Datasets

We selected two curated datasets from the DIABIMMUNE project and one Seeding dataset obtained from (Song et al., 2021) . The DIABIMMUNE project aims to investigate the relationship between the human microbiome and the development of autoimmune diseases. The curated datasets from DIABIMMUNE provide a valuable resource for studying the dynamics of the microbiome over time (Kostic et al., 2015; Vatanen et al., 2016; Yassour et al., 2016) . The Seeding dataset, on the other hand, offers an opportunity to evaluate the performance of our method on a different type of microbiome data, ensuring a comprehensive assessment of MNLVAE.

### 3.2. Data Preprocessing

Before conducting our experiments, we preprocessed the datasets to ensure compatibility with our method. This preprocessing step involved filtering the datasets and splitting them into training and testing sets, enabling us to assess the prediction performance of our method on unseen data. Additionally, we utilized a validation dataset for MNLVAE, which was derived from the initial training dataset, to prevent overfitting and to fine-tune the model's hyperparameters by early stopping.

In the data preprocessing phase, we encountered a challenge with the Seeding and Diabimmune-I datasets when using gLV, which employs ALR as a transformation. In order to calculate the log-ratio, it was necessary to have at least one microbe counted in all observations to serve as a denominator. Consequently, we removed some observations from these datasets to satisfy this requirement. For the MNLVAE experiments, we denoted these modified datasets using the abbreviation $(.)_{gLV}$.

### 3.3. Experimental Setup

For our experiments, we compared the performance of MNLVAE with the gLV model. We evaluated the prediction performance of the models using the normalized root mean squared error (NRMSE) metric, focusing on test subjects and test observations that are common across all datasets, specifically those based on the generalized Lotka-Volterra (gLV) model. Additionally, we investigated the impact of different GP configurations, data dimensions, and data transformations on the performance of MNLVAE. This comprehensive evaluation allowed us to assess the effectiveness of our method across a wide range of scenarios and condi-

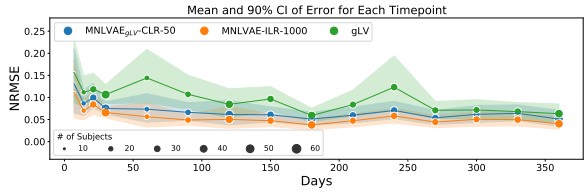

(a) Seeding Dataset

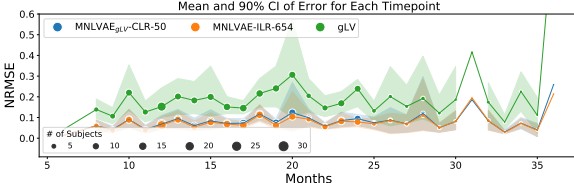

(b) DIABIMMUNE-I Dataset

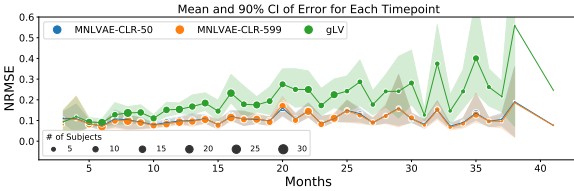

(c) DIABIMMUNE-II Dataset

*Figure 2.* Comparison of NRMSE over time for the test subjects' unobserved measurements. The subfigures illustrate the performance of the models over three datasets.

tions. In our analysis, we explored the performance of the MNLVAE model under various data dimensionalities. We selected four different dimensionalities for this purpose according to sparsity of the microbe counts: A dimensionality of 50, which is a common choice for reduced data representation and the one used in the gLV model. A dimensionality of 300, allowing us to investigate the model's performance at a higher level of granularity. The maximum available dimensions for the Diabimmune datasets, which are 654 and 599, representing the complete set of dimensions for each respective dataset. A dimensionality of 1000 for the Seeding dataset, as a higher-dimensional choice to assess the model's behavior under a large number of dimensions.

In our experiments, we utilized a single hidden layer for both the encoder and decoder networks. We also tested various different latent dimensions and selected the best performing one for our final results. The detailed neural network structures are given in Supplementary Material-A.

### 3.4. Results

We observed that MNLVAE consistently outperforms alternative approaches across all datasets, as illustrated in Fig. 2. As shown in Table 1 for DIABIMMUNE datasets, the performance differences between various GP configurations

*Table 1.* Comparison of the GP Configurations for the two datasets, DIABIMMUNE-I and DIABIMMUNE-II. GP Conf. abbreviations: T = time, ID = individual identifier, T1D = type 1 diabetes status, G = gender, and C = country.

|  | DIABIMMUNE-I | | DIABIMMUNE-II | |
|---|---|---|---|---|
| GP Conf | $\text{CLR}_{gLV}$-50 | ILR-654 | CLR-50 | CLR-599 |
| T+ID | 0.138 | 0.128 | 0.139 | 0.136 |
| T+ID+T1D | **0.132** | **0.125** | 0.134 | **0.129** |
| T+ID+G | **0.132** | **0.125** | 0.134 | 0.131 |
| T+ID+G+T1D | 0.133 | 0.126 | **0.132** | 0.130 |
| T+ID+G+C | 0.136 | 0.126 | 0.133 | 0.131 |
| T+ID+G+C+T1D | 0.139 | **0.125** | 0.134 | 0.131 |

*Table 2.* NRMSE values for each model in the respective datasets

| MODEL | SEEDING | DIABIMMUNE-1 | DIABIMMUNE-2 |
|---|---|---|---|
| **MNLVAE** | | | |
| ILR-MAX | **0.093** | **0.123** | 0.131 |
| CLR-MAX | 0.094 | 0.124 | **0.128** |
| ILR-300 | 0.098 | 0.124 | 0.129 |
| CLR-300 | 0.095 | 0.125 | 0.130 |
| ILR-50 | 0.100 | 0.130 | 0.132 |
| CLR-50 | 0.105 | 0.132 | 0.130 |
| $\text{ILR}_{\text{GLV}}$-50 | 0.108 | 0.132 | - |
| $\text{CLR}_{\text{GLV}}$-50 | 0.108 | 0.130 | - |
| **GLV** | | | |
| GLV-50 | 0.164 | 0.304 | 0.277 |

in MNLVAE were not significant in most cases, indicating that our model is robust to the choice of GP configurations and serves as a versatile tool for microbiome time-series analysis. We also noted that different datasets display distinct behavior, underscoring the importance of evaluating our method on a diverse range of microbiome time-series data to fully comprehend its capabilities and limitations.

Our experiments demonstrated that increased data dimensions consistently result in enhanced prediction performance for MNLVAE, as shown in Table 2. This observation highlights the advantages of utilizing the high-dimensional nature of microbiome data, allowing our model to more effectively capture complex interactions and dependencies among microbial taxa. Table 2 also shows the impact of CLR and ILR transformations on the performance of MN-LVAE. Our results indicate that for lower data dimensions, the CLR transformation yields better performance, while for higher data dimensions, the ILR transformation outperforms CLR. This insight can help guide the choice of data transformation techniques in future applications of our method.

We also explored the effect of including a key covariate, such as birth mode or T1D status, on MNLVAE's prediction performance. Our results suggest that in simpler GP configurations, the addition of a key covariate may have a positive impact on performance. However, further investigation is needed to fully understand the role of indicator covariates in our model and to determine the best strategy

for incorporating covariates into MNLVAE.

## 4. Discussion

In this paper, we presented MNLVAE, a novel method for analyzing compositional longitudinal data, specifically designed for microbiome time-series analysis. Our experiments have demonstrated that MNLVAE outperforms competing methods and offers improved prediction performance across different datasets. Nevertheless, it is not without limitations.

While our MNLVAE model employs interactive kernels in the Gaussian process prior to capture some forms of non-additive interactions among microbial taxa, it is worth noting that these may not fully represent all possible complex interactions.

We have addressed the scalability issue with an efficient, mini-batch compatible upper bound for the KL divergence, which assumes a low-rank inducing point approximation for the covariance function. While this solution enhances computational efficiency, it may introduce bias or affect the model's ability to fully capture the data's complexity.

Future research should focus on developing strategies on exploring extensions to our model that can incorporate a wider range of complex interactions. Additionally, further investigation into the impact of our assumption of a low-rank inducing point approximation on the model's performance, and potential alternatives to this approximation, could be beneficial.

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

# A. Supplementary Material

## A.1. Experimental Setup

|  | Hyperparameter | Seeding | DIABIMMUNE-I | DIABIMMUNE-II |
|---|---|---|---|---|
| Inference network | Dimensionality of input | 50, 300, 1000 | 50, 300, 654 | 50, 300, 599 |
|  | Number of feedforward layers | 1 | 1 | 1 |
|  | Number of elements in each feedforward layer | 15 | 15 | 15 |
|  | Dimensionality of latent space | 18,20,22,24 | 6,8,10,12,18 | 6,8,10,12,18 |
|  | Activation function of layers | SoftPlus | SoftPlus | SoftPlus |
| Generative network | Dimensionality of input | 18,20,22,24 | 6,8,10,12,18 | 6,8,10,12,18 |
|  | Number of feedforward layers | 1 | 1 | 1 |
|  | Number of elements in each feedforward layer | 15 | 15 | 15 |
|  | Activation function of feedforward layers | SoftPlus | SoftPlus | SoftPlus |
|  | Transformations | inv ILR, Softmax | ILR, Softmax | ILR, Softmax |

*Table 3.* Neural network architectures used in the Seeding, DIABIMMUNE-I, and DIABIMMUNE-II datasets.

