# OpenReview forum: "Longitudinal Variational Autoencoder for Compositional Data Analysis"
_ICML.cc/2023/Workshop/IMLH — IMLH 2023 PosterShortPaper_

### Official Review · Reviewer_VB8i · 2023-06-10
**The analysis of compositional longitudinal data, particularly in the context of microbiome time-series, is an important and actively researched area. The proposed MNLVAE model offers a promising approach to address the challenges associated with analyzing high-dimensional, sparse, and compositional count data. The experiments conducted on real-world microbiome datasets demonstrate the superiority of MNLVAE over existing approaches in terms of predictive capabilities. The findings of this work have the potential to advance the understanding of microbial community dynamics and contribute to the field of microbiome research.**

**Rating:** 8
**Confidence:** 3

**Review:**

Pros:

Introduction of a novel generative model, MNLVAE, for analyzing compositional longitudinal data in microbiome time-series.


Cons:

Some parts of the paper could benefit from further clarification, particularly in terms of notation and experimental setup details.
The paper lacks a discussion of the limitations of the proposed method and potential future directions for research.

Overall, the work is of high quality and provides a significant contribution to the field of microbiome time-series analysis.

---

### Official Review · Reviewer_TABm · 2023-06-19
**The paper deals with an important problem in microbiome analysis and proposes a novel generative model**

**Rating:** 6
**Confidence:** 3

**Review:**

The paper proposes a novel approach MNLVAE to analyze compositional longitudinal data, specifically in microbiome time-series. The MNLVAE model utilizes a Gaussian process (GP) prior and deep learning techniques to capture complex interactions among microbial taxa while considering the compositional structure of the data.

The proposed generative model is novel and the paper is well-written. The largest concern about the paper is the lack of comparison methods and make the evaluation to be insufficient - the gLV model is the only comparison method in the current paper. There are different types of available models for compositional time-series analysis (e.g. dynamic linear models, RNN models, Dirichlet-Multinomial model, temporal probabilistic model, etc.). It would be highly desirable if a comprehensive comparison between MNLVAE and other methods can be made to better evaluate the proposed method.

---

### Meta-Review · Area_Chair_Hh81 · 2023-06-18

**Recommendation:** Accept (Poster)
**Confidence:** 3

**Metareview:**

The paper presents a novel generative model, MNLVAE, for analyzing compositional longitudinal data in microbiome time-series.

All reviews recommend acceptance and recognize the contribution of the paper.

---

### Decision · Program_Chairs · 2023-06-20

Accept (Poster Short Paper)